# Genomic risk score offers predictive performance comparable to clinical risk factors for ischaemic stroke

Gad Abraham[1,2,3]*, Rainer Malik[4], Ekaterina Yonova-Doing[5], Agus Salim[6,7], Tingting Wang[1], John Danesh[5,8,9,10,11,12], Adam S. Butterworth [5,8,9,10,11,12], Joanna M.M. Howson [5,11], Michael Inouye[1,2,3,5,11,13,16]* & Martin Dichgans [4,14,15,16]*

Recent genome-wide association studies in stroke have enabled the generation of genomic risk scores (GRS) but their predictive power has been modest compared to established stroke risk factors. Here, using a meta-scoring approach, we develop a metaGRS for ischaemic stroke (IS) and analyse this score in the UK Biobank ($n = 395,393$; 3075 IS events by age 75). The metaGRS hazard ratio for IS (1.26, 95% CI 1.22–1.31 per metaGRS standard deviation) doubles that of a previous GRS, identifying a subset of individuals at monogenic levels of risk: the top 0.25% of metaGRS have three-fold risk of IS. The metaGRS is similarly or more predictive compared to several risk factors, such as family history, blood pressure, body mass index, and smoking. We estimate the reductions needed in modifiable risk factors for individuals with different levels of genomic risk and suggest that, for individuals with high metaGRS, achieving risk factor levels recommended by current guidelines may be insufficient to mitigate risk.

[1] Cambridge Baker Systems Genomics Initiative, Baker Heart and Diabetes Institute, Melbourne, VIC, Australia. [2] Cambridge Baker Systems Genomics Initiative, Department of Public Health and Primary Care, University of Cambridge, Cambridge, UK. [3] Department of Clinical Pathology, University of Melbourne, Parkville, VIC, Australia. [4] Institute for Stroke and Dementia Research, University Hospital, Ludwig-Maximilians-Universität LMU, Munich, Germany. [5] British Heart Foundation Cardiovascular Epidemiology Unit, Department of Public Health and Primary Care, University of Cambridge, Cambridge, UK. [6] Baker Heart and Diabetes Institute, Melbourne, VIC, Australia. [7] Department of Mathematics and Statistics, La Trobe University, Melbourne, VIC, Australia. [8] British Heart Foundation Centre of Research Excellence, University of Cambridge, Cambridge, UK. [9] National Institute for Health Research Blood and Transplant Research Unit in Donor Health and Genomics, University of Cambridge, Cambridge, UK. [10] National Institute for Health Research Cambridge Biomedical Research Centre, University of Cambridge and Cambridge University Hospitals, Cambridge, UK. [11] Health Data Research UK Cambridge, Wellcome Genome Campus and University of Cambridge, Cambridge, UK. [12] Department of Human Genetics, Wellcome Sanger Institute, Hinxton, UK. [13] The Alan Turing Institute, London, UK. [14] German Center for Neurodegenerative Diseases (DZNE), Munich, Germany. [15] Munich Cluster for Systems Neurology (SyNergy), Munich, Germany. [16] These authors jointly supervised: Michael Inouye, Martin Dichgans. *email: gad.abraham@baker.edu.au; mi336@medschl.cam.ac.uk; martin.dichgans@med.uni-muenchen.de

Stroke is a leading cause of death worldwide and the leading cause of permanent disability[1,2]. About 80% of stroke cases are of ischaemic origin[3]. The risk of ischaemic stroke (IS) is determined by a complex interplay of genetic and environmental factors partly acting through modifiable risk factors such as hypertension and diabetes. Roughly thirty-five genomic loci have been robustly associated with stroke[4–7], and many more genetic associations have been reported for stroke-related risk factors[8–14], e.g., over 1000 loci have been associated with blood pressure (BP)[11,15–19] and >100 with atrial fibrillation (AF)[10,20]. These data are now beginning to be harnessed to aid risk prediction.

Recent work has highlighted the potential of genomic risk scores (GRS) for risk prediction of common diseases[21–24]. Genomic risk prediction has a notable advantage over established risk factors as it could be used to infer risk of disease from birth, thus allowing the initiation of preventive strategies before conventional risk factors manifest and their discriminative capacity begins to emerge.

For stroke, a recent 90-SNP GRS derived from the MEGA-STROKE GWAS meta-analysis[4] showed that genetic and lifestyle factors are independently associated with incident stroke[24], and that even among individuals with high GRS, lifestyle factors had a large impact on risk, implying that risk could be reduced in those with high genetic predisposition for stroke. However, in contrast to GRSs for other cardiovascular diseases like coronary artery disease (CAD)[21–23], the predictive power of previous GRS for stroke has been limited[25–27], likely because of limited genetic data for stroke and the well-known heterogeneity of the stroke phenotype[4,7]. Recent analytical advances have enabled more powerful GRS construction, such as those leveraging multiple sets of GWAS summary statistics[21,28], potentially allowing for power and heterogeneity limitations to be overcome. Specifically, for CAD, an approach where multiple GRSs are combined into one meta-score (metaGRS) was found to improve risk prediction over any one of the individual CAD GRS[21]. Such an approach may be widened to provide substantively improved genomic prediction of stroke.

Here, we extend the metaGRS strategy to predict IS by incorporating GWAS summary statistics for stroke and its aetiological subtypes along with GWAS summary statistics for risk factors and co-morbidities of IS. This new IS metaGRS is validated and compared with previously published GRS using UK Biobank[29,30]. We next compare the predictive capacity of the IS metaGRS to established non-genetic risk factors for IS. Finally, we assess the additional information provided by the metaGRS in combination with current guidelines for the treatment of established IS risk factors and create joint models which predict absolute risk of incident IS.

## Results

**Derivation of a metaGRS for ischaemic stroke.** To create the GRSs we randomly split the UK Biobank (UKB) British white dataset ($n = 407,388$) into a derivation ($n = 11,995$) and validation set ($n = 395,393$; "Methods" section Fig. 1, Table 1). In order to increase statistical power in the derivation phase, we enriched the derivation set ($n = 11,995$) with IS events ($n = 888$, 7.4%). A schematic of the overall study design is given in Fig. 1.

We used GWAS summary statistics that did not include the UKB for five stroke outcomes and 14 stroke-related phenotypes (Supplementary Table 1) to generate 19 GRSs associated with IS (Fig. 1). As expected, the 19 individual GRSs were correlated with each other in several distinct clusters: (i) any stroke (AS), IS, cardioembolic stroke (CES), large artery stroke (LAS), and small vessel stroke (SVS); (ii) the three CAD scores (1KGCAD, 46K,

and FDR202); (iii) total cholesterol (TC), triglycerides (TG), low-density lipoprotein cholesterol (LDL), and high-density lipoprotein cholesterol (HDL); (iv) systolic BP (SBP) and diastolic BP (DBP); and (v) body mass index (BMI) and type 2 diabetes (T2D) (Fig. 2). From the 19 distinct GRSs, we constructed the metaGRS using elastic-net logistic regression with 10-fold cross-validation on the derivation set (Fig. 1; metaGRS; model weights are shown in Supplementary Fig. 1), and subsequently converted the model to a set of 3.2 million SNP weights, which are freely available (https://doi.org/10.6084/m9.figshare.8202233).

We performed a sensitivity analysis to assess whether the estimation of the metaGRS weights on the UKB derivation set led to over-fitting (upwards bias in apparent performance) of the score in the validation set. We developed a metaGRS based on four component GRSs (AS, IS, BMI, and SBP) in cross-validation on the derivation set. We compared this metaGRS with a score derived using smtPred[28], which relies on the chip heritabilities and genetic correlations estimated from the GWAS summary statistics via LD score regression[31,32], independently of the UKB (Supplementary Fig. 2). Overall, the two scores were highly correlated (Pearson $r = 0.98$), and had indistinguishable associations with IS in the UKB validation set, indicating that our metaGRS procedure did not lead to overfitting in the validation set.

**The metaGRS improves risk prediction of ischaemic stroke compared with other genetic scores.** Using the independent UKB validation set, we next quantified the risk prediction performance of the metaGRS, and evaluated its association with IS via survival analysis. The metaGRS was associated with IS with a hazard ratio (HR) of 1.26 (95% CI 1.22–1.31) per standard deviation of metaGRS, which was stronger than any individual GRS comprising the metaGRS (including the IS GRS [HR = 1.18, 95% CI 1.15–1.22]) and was twice the effect size of the previously published 90-SNP IS score[24] (HR = 1.13 [95% CI 1.10–1.17]; Supplementary Fig. 3a). The metaGRS also increased the C-index by 0.029 over the 90-SNP GRS (Supplementary Fig. 3b). We also assessed the performance of the IS metaGRS for predicting the AS outcome. We found the associations were consistently weaker for AS than for IS, however, as with IS, the metaGRS was a stronger predictor of AS than the 90-SNP GRS score (Supplementary Fig. 3).

In a Kaplan–Meier analysis of IS, the top and bottom 10% of the metaGRS showed substantial differences in cumulative incidence of IS (Supplementary Fig. 4; log-rank test between the top decile and the 45–55% decile: $P = 3 \times 10^{-6}$); these results were consistent with a Cox proportional hazards model of the metaGRS assessing the HRs for the top 10% decile vs the middle 45–55% decile (Supplementary Fig. 5). The top 0.25% of the population were at a threefold increased risk of IS vs. the middle decile (45–55%), with HR = 3.0 (95% CI 1.96–4.59) (Fig. 3).

There was no evidence for a statistical interaction of the metaGRS with sex on IS hazard (Wald test in Cox proportional hazard model, $P = 0.614$), indicating that the substantial differences in cumulative incidence between the sexes were driven by differences in baseline hazards rather than by any sex-specific effects of the metaGRS itself.

A small number of individuals ($n = 45$) had recorded haemorrhagic stroke before their primary IS event. We conducted two sensitivity analyses to assess the impact of this on our results: (i) excluding $n = 45$ individuals from the analysis; (ii) adjusting for haemorrhagic stroke status in the analysis. In both cases, there was essentially no difference in the association of the metaGRS with IS compared with the original analysis (HR = 1.27 per standard deviation of the metaGRS across the two analyses).

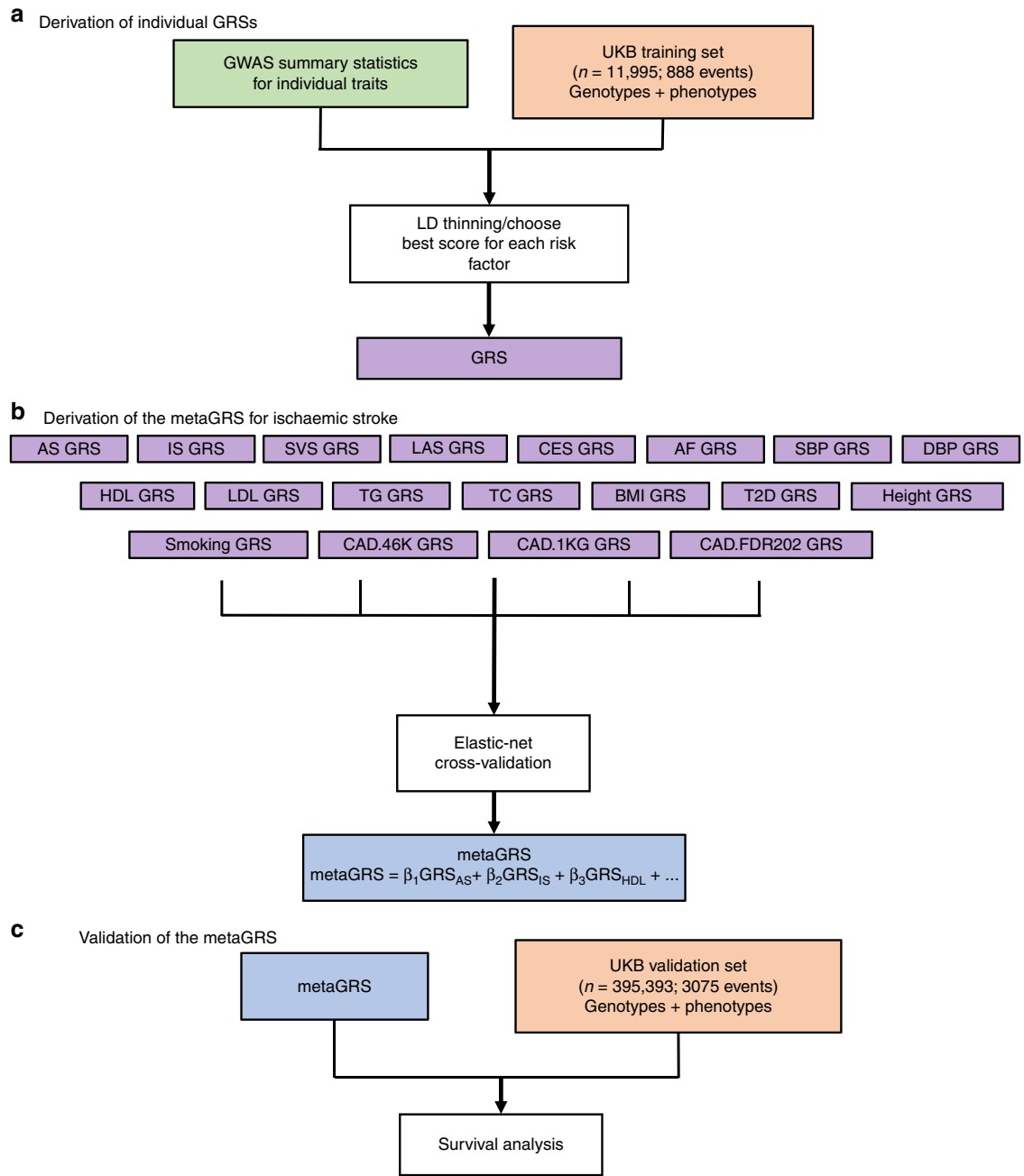

**Fig. 1 Study design. a** Individual GRSs were derived in the UK Biobank training set ($n = 11{,}995$) using GWAS summary statistics for individual traits. **b** The metaGRS for ischaemic stroke was then derived by integrating individual GRSs using elastic-net cross-validation. **c** Validation of the metaGRS for ischaemic stroke was performed in the UK Biobank validation set ($n = 395{,}393$). UKB UK Biobank, GWAS genome-wide association study, GRS genomic risk score.

To further assess the contribution of different classes of GRS in the final score, we constructed two metaGRSs: (i) a score based on stroke-related GRSs (AS, IS, CES, LAS, SVS) and CAD-related GRSs (46K, FDR202, 1KGCAD) but no other risk factors; and (ii) a metaGRS based on stroke-related GRSs (AS, IS, CES, LAS, SVS) and risk-factor-related GRSs (SBP, DBP, TC, LDL, HDL, TG, AF, BMI, Height, T2D, Smoking) but no CAD-related GRSs (Supplementary Fig. 6). Addition of either risk-factor GRSs or CAD-related GRSs each led to more powerful metaGRSs compared with the IS-only GRS, but the best score was achieved when combining both types of GRS into the 19-component metaGRS, indicating that both types of GRS had independent information about stroke risk. Note that due to pleiotropy there is some overlap between the genetic signal for CAD and risk factors such as BP and cholesterol.

**The ischaemic stroke metaGRS has comparable or higher predictive power than established risk factors**. We next compared the performance of the metaGRS with established risk factors[33] for predicting IS. We examined seven risk factors at the first UKB assessment: LDL cholesterol, SBP, family history of stroke, BMI, diabetes diagnosed by a doctor, current smoking, and hypertension (an expanded definition based on SBP/DBP measurements, BP medication usage, self-reporting, and hospital records; "Methods" section).

**Table 1 Study characteristics of the UK Biobank validation dataset.**

| Baseline characteristic | UK Biobank N = 395,393 | Male N = 180,653 (45.7%) | Female N = 214,740 (54.3%) |
|---|---|---|---|
| Age, years [mean (sd)] | 56.9 (8.0) | 57.1 (8.1) | 56.7 (7.9) |
| Current smoker, N (%) | 39,804 (10.0%) | 21,261 (11.8%) | 18,543 (8.6%) |
| Systolic blood pressure, mm Hg [mean (sd)] (adjusted for BP medication) | 143.3 (21.7) | 146.9 (20.4) | 140.2 (22.2) |
| Diabetes diagnosed by doctor, N (%) | 18,675 (4.7%) | 11,449 (6.3%) | 7226 (3.4%) |
| Hypertension, N (%) | 211,069 (53.4%) | 110,540 (61.2%) | 100,529 (46.8%) |
| Family history of stroke, 1st degree relative, N (%) | 104,831 (26.5%) | 45,569 (25.2%) | 59,262 (27.6%) |
| High cholesterol, N (%) | 53,141 (13.4%) | 30,670 (17.0%) | 22,471 (10.5%) |
| Prevalent stroke events, N (%), any stroke before age 75 | 4543 (1.1%) | 2679 (1.5%) | 1864 (0.9%) |
| Prevalent stroke events, N (%), ischaemic stroke before age 75 | 1152 (0.3%) | 787 (0.4%) | 365 (0.2%) |
| Incident stroke events, N (%), any stroke before age 75 | 2607 (0.7%) | 1531 (0.8%) | 1076 (0.5%) |
| Incident stroke events, N (%), ischaemic stroke before age 75 | 1923 (0.5%) | 1207 (0.7%) | 716 (0.3%) |
| On blood-pressure lowering medication, N (%) | 80,880 (20.5%) | 43,714 (24.2%) | 37,166 (17.3%) |
| On lipid-lowering medication, N (%) | 66,739 (16.9%) | 40,164 (22.2%) | 26,575 (12.4%) |
| Follow-up time, years [mean (sd)] | 6.3 (1.9) | 6.2 (2.1) | 6.4 (1.8) |

Shown are characteristics obtained at the first UK Biobank assessment

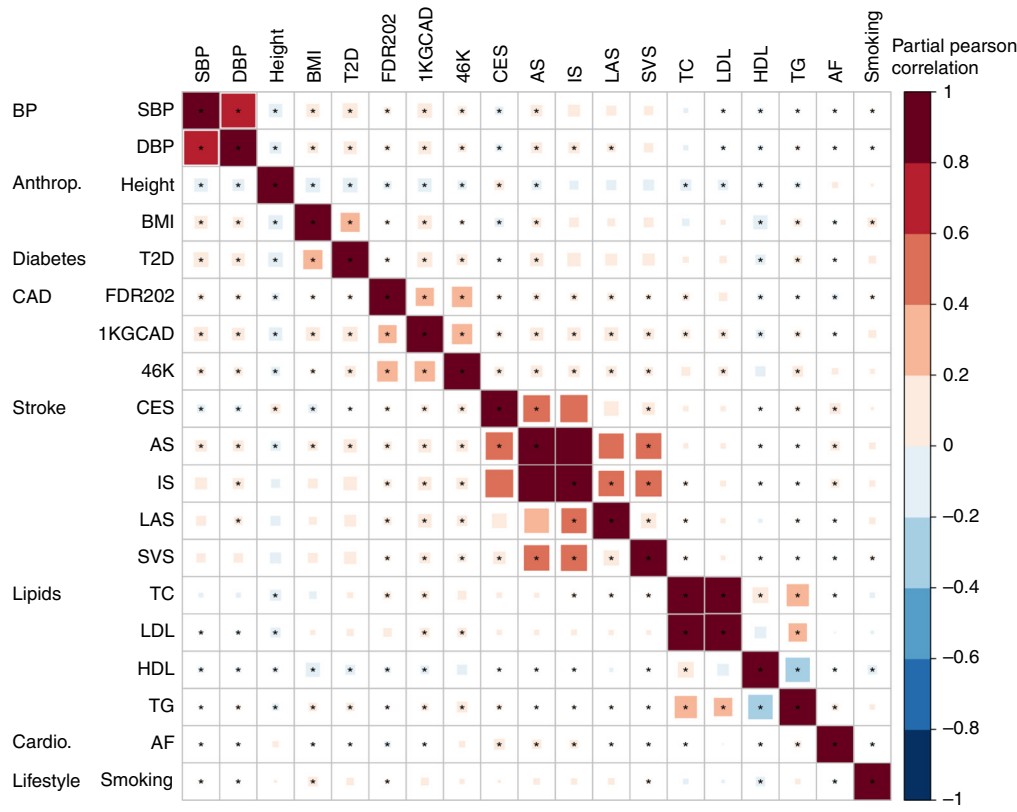

**Fig. 2 Individual GRSs for stroke-related phenotypes and stroke outcomes correlate in several distinct clusters.** Shown is the partial Pearson correlation plot of individual GRSs in a random sample of 20,000 UK Biobank individuals. Estimates are from linear regression of each pair of standardised GRSs, adjusting for genotyping chip (UKB/BiLEVE) and 10 PCs. Stars indicate Benjamini–Hochberg false discovery rate < 0.05 (adjusting for 171 tests). GRSs were ordered via hierarchical clustering of the absolute correlation. *Anthrop* anthropometric, *cardio* cardiovascular (other than CAD), *SBP* systolic blood pressure, *DBP* diastolic blood pressure, *Height* measured height, *BMI* body mass index, *T2D* type 2 diabetes, *1KGCAD* coronary artery disease from 1000 Genomes, *46K* coronary artery disease from Metabochip, *FDR202* coronary artery disease from 1000 Genomes (top SNPs), *CES* cardioembolic stroke, *AS* any stroke, *IS* ischaemic stroke, *LAS* large artery stroke, *SVS* small vessel stroke, *TC* total cholesterol, *LDL* low-density lipoprotein cholesterol, *HDL* high-density lipoprotein cholesterol, *TG* triglycerides, *AF* atrial fibrillation, *Smoking* cigarettes per day.

As expected, established risk factors were positively associated with incident IS, with hypertension being the strongest risk factor (Supplementary Fig. 7). Notably, the HR of the metaGRS (incident IS HR = 1.25 per s.d.) was similar to that of SBP (incident IS HR =

1.28 per s.d., where the s.d. of SBP was 21.7 mm Hg) and current smoking (incident IS HR = 1.25, s.d. = 0.3) (Supplementary Fig. 7).

Comparison of the C-index for time to incident IS revealed that BP phenotypes, hypertension and SBP (C = 0.590 [95% CI

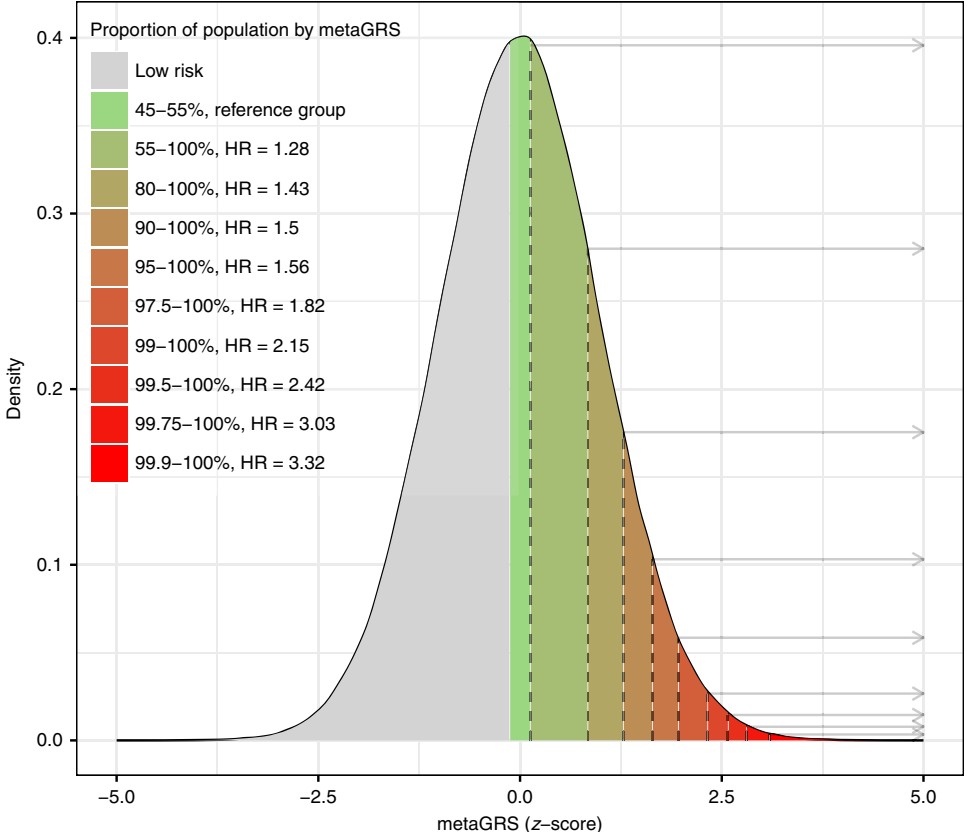

**Fig. 3 The metaGRS identifies individuals at increased risk of ischaemic stroke.** Shown is the distribution of the metaGRS for ischaemic stroke in the UK Biobank validation set ($n = 395,393$), and corresponding hazard ratios. Hazard ratios are for the top metaGRS bins (stratified by percentiles) vs. the middle metaGRS bin (45–55%).

0.577–0.603]; C = 0.584 [95% CI 0.570–0.598], respectively), had the largest C-indices, followed by the metaGRS (C = 0.580 [95% CI 0.566–0.593]) and the remaining established risk factors (Fig. 4). Notably, the metaGRS had a greater C-index than family history of stroke (C = 0.558, 95% CI 0.544–0.572; Fig. 4). The metaGRS and hypertension contained similar additional information on top of the other risk factors; adding either the metaGRS or hypertension to the six other risk factors yielded similar predictive power, C = 0.629 (95% CI 0.615–0.643) and C = 0.628 (95% 0.614–0.641), respectively. Finally, adding both the metaGRS and hypertension to the six risk factors yielded the model with the highest C-index, C = 0.637 (95% CI 0.623–0.650) (Fig. 4). Note that LDL-cholesterol was not included in this analysis as it had only weak associations with stroke and is not considered a major stroke risk factor.

**The metaGRS contributes to ischaemic stroke risk independent of established risk factors.** Given that the metaGRS is composed of GRSs for stroke and stroke risk factors, we conducted several complementary analyses to assess the association of the metaGRS with these risk factors, and whether the metaGRS was associated with IS risk independently of these risk factors. As expected, the IS metaGRS was positively and significantly associated with all seven risk factors (Supplementary Table 2). Adjusting for these risk factors as well as BP-lowering and/or lipid-lowering medication status only modestly attenuated the association of the metaGRS with incident IS (Supplementary Fig. 8), indicating that the information contained in the metaGRS was only partially explained by these factors. On the other hand, adjusting for the metaGRS modestly but consistently attenuated the association of each risk factor itself with IS risk (Supplementary Fig. 7). There

was no evidence for statistical interaction of the metaGRS effects on incident IS with medication status at assessment (Wald test in logistic regression, $P = 0.23$ and $P = 0.82$ for interaction of the metaGRS with BP medication and cholesterol-lowering medication, respectively).

**Predicting ischaemic stroke risk with established risk factors and the metaGRS.** The clinical utility of a GRS depends on its performance in combination with established risk factors and risk models. To examine this, we conducted analyses integrating information on risk factor levels based on (i) recent ACC/AHA/AAPA/ABC/ACPM/AGS/APhA/ASH/ASPC/NMA/PCNA guidelines[34] (SBP < 120 mm Hg); (ii) AHA/ASA guidelines for primary prevention of stroke[33] (BMI < 25 kg m$^{-2}$); (iii) smoking status and diabetes status. We used Cox models of these established risk factors and the metaGRS together with the estimated baseline cumulative hazards to predict cumulative incidence of IS for individuals with a high metaGRS (top 1%), average metaGRS (50%), and low metaGRS (bottom 1%) along with two levels of risk factors: (i) meeting guideline targets for the above risk factors[34] and (ii) the following combination of risk factors representative of an individual at typical stroke risk: SBP = 140 mm Hg, BMI = 30 kg m$^{-2}$, current smoking, and no diagnosed diabetes.

The predicted risk of IS for individuals with a high metaGRS (top 1%) and high levels of risk factors was maximal by age 75, reaching a cumulative incidence of 8.5% (95% CI 5.2–11.6%) for males and 5.1% (95% CI 3.1–7.1%) for females (Fig. 5a). Effective reduction in the levels of the modifiable risk factors (SBP, BMI, and smoking) to match guideline targets was predicted to result in a substantial reduction in risk, down to 2.8% (95% CI

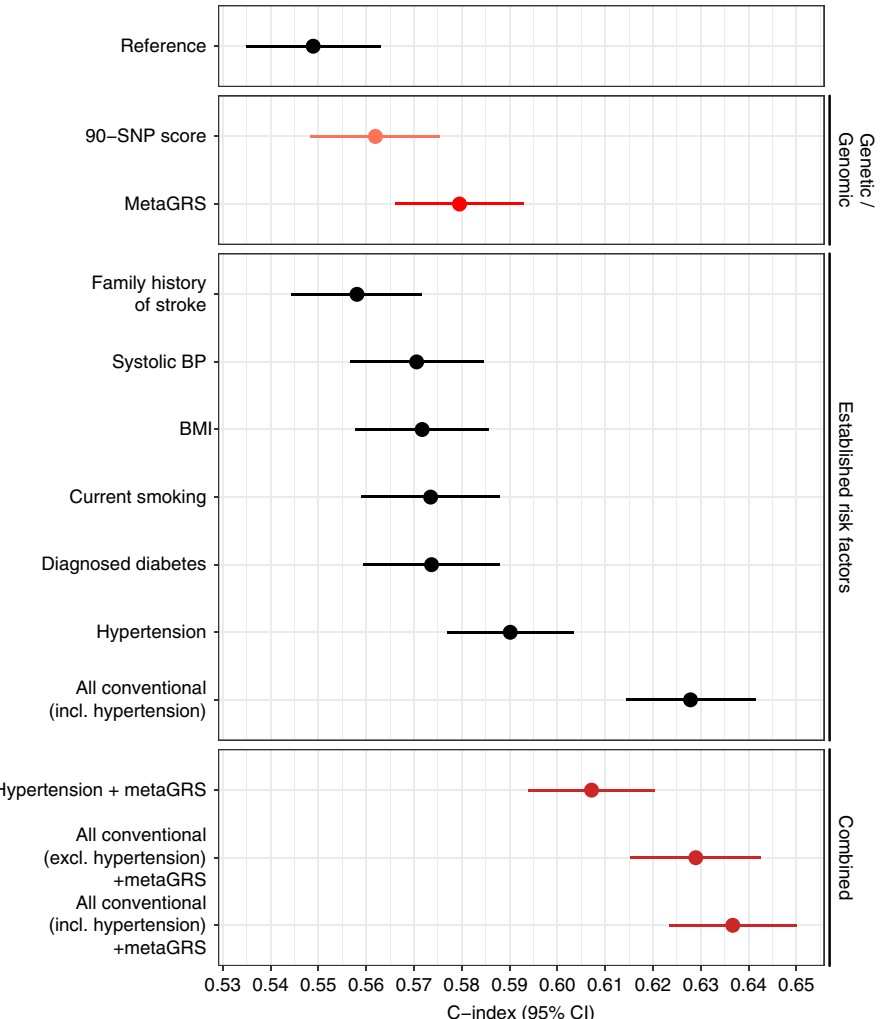

**Fig. 4 The metaGRS for ischaemic stroke has comparable or higher predictive power than established risk factors.** Shown are the C-indices for incident stroke in the UKB validation set comparing the metaGRS with established risk factors. The reference model included the genotyping chip and 10 genetic PCs. Results are for the UKB validation set, excluding prevalent stroke events (n = 390,849). Red circles represent genetic/genomic scores; black circles represent non-genetic scores. Error bars represent 95% confidence intervals.

1.7–3.9%) for males and 1.7% (95% CI 1.0–2.4%) for females by age 75, thus substantially compensating for the high genomic risk.

Conversely, for individuals matching the guidelines for established risk factors (Fig. 5b), there were notable differences in IS incidence for individuals in the top (1%) compared with the bottom (1%) of the metaGRS; with 2.8% (95% CI 1.7–3.9%) vs. 1.2% (95% CI 0.7–1.7%) in males and 1.7% (95% CI 1.0–2.4%) vs. 0.7% (95% CI 0.4–1.0%) in females, respectively, by age 75. These results further indicate that the metaGRS captures residual risk of stroke not quantified by existing risk factors.

## Discussion
In this study, we developed a genomic risk score for IS based on GWAS summary statistics for 19 stroke and stroke-related traits. We quantify the predictive power of the IS metaGRS by comparing it to previously published genetic scores and measures of established non-genetic risk factors, and demonstrate its added value in combination with established risk factors and in the context of current guidelines for primary stroke prevention. While GRS for stroke are not yet at the level necessary for clinical translation, our analyses constitute several significant advances.

First, we showed that the IS metaGRS had stronger association with IS than previously published genetic scores, doubling the effect size of the most recent genetic score. To put its performance in context, we estimated the IS metaGRS identified the 1 in 400 individuals who were at threefold increased risk of IS, a level of risk and frequency similar to common monogenic cardiovascular diseases, such as familial hypercholesterolaemia (FH), a risk factor for myocardial infarction[35]. Monogenic forms of stroke, such as CADASIL, are relatively rare[36], thus the IS metaGRS may represent a potential new avenue to more common polygenic risk stratification, in combination with established risk factors.

Second, the IS metaGRS had comparable predictive power to SBP, higher predictive power than other established risk factors measured, apart from hypertension, and captured residual risk not quantified by the established risk factors. In anticipation of a potential role in early screening, we estimate the risk reduction through modifiable stroke risk factors across different metaGRS backgrounds, and further show that current guidelines for stroke risk factors may be insufficiently stringent for individuals at high metaGRS.

Third, we explicitly modelled how changes in modifiable risk factors, such as SBP and BMI, can compensate for high genomic risk. Previous research has demonstrated that intervening on

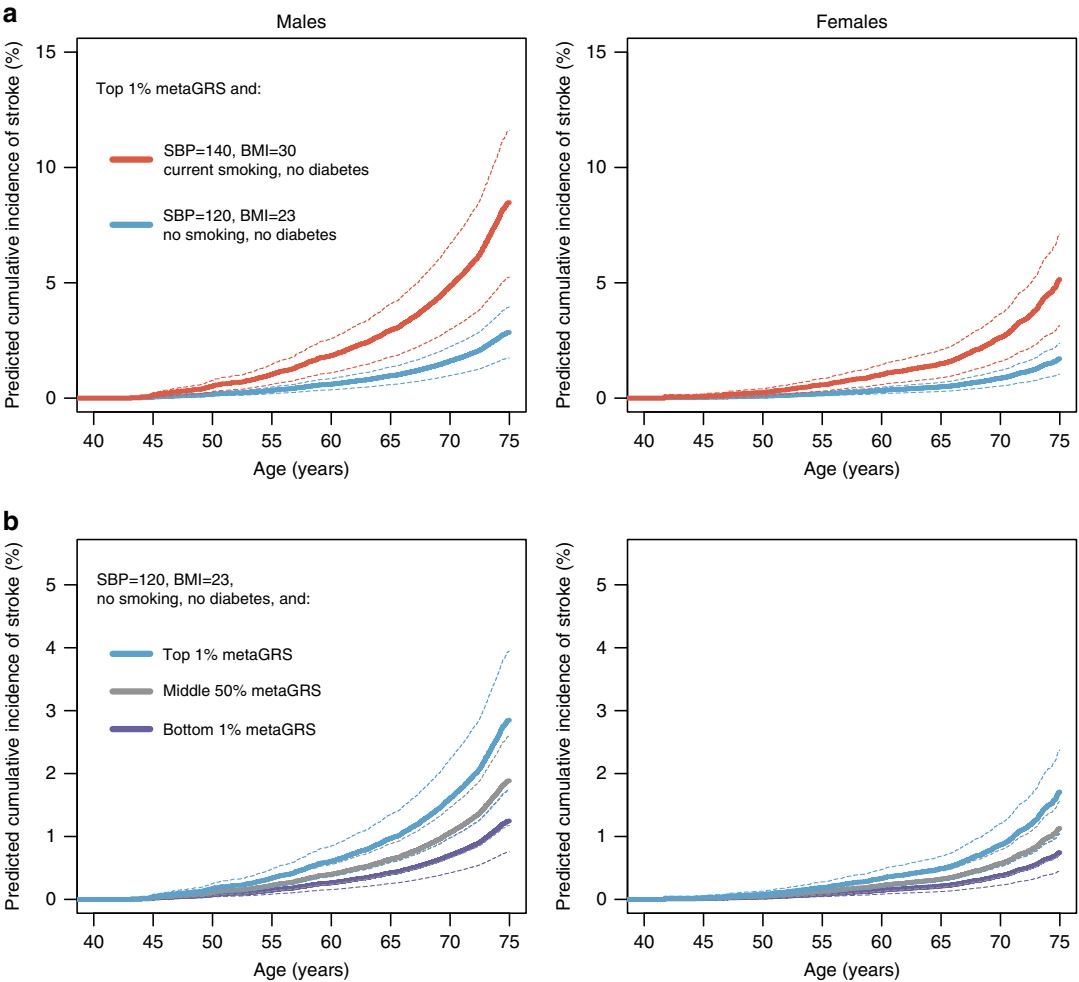

**Fig. 5 Predicted cumulative incidence of ischaemic stroke.** Shown is the predicted cumulative incidence of IS in subjects with either (**a**) high levels of the metaGRS along with different risk factor levels (red: outside the guidelines; cyan: within the guidelines); or (**b**) risk factors within accepted guidelines along with different levels of the metaGRS (cyan: top 1% of the metaGRS; grey: middle 50% of the metaGRS; dark blue: bottom 1% of the metaGRS). Results are based on the UKB validation set, excluding prevalent stroke events (n = 390,849). Error bars represent 95% confidence intervals.

modifiable risk factors can compensate for increased genetic risk of disease[21,37]. However, these analyses relied on simply counting the number of elevated risk factors, which does not account for the differences in effect size between various risk factors. Importantly, our approach was flexible in that various combinations of risk factor reductions can lead to the same outcome in terms of risk.

Our approach shows, for different genomic risk backgrounds, how modifiable risk factors could, in principle, be tailored to an individual's ability to reduce an established risk factor(s) while maintaining an overall acceptable level of absolute risk. Similarly, this approach could potentially be used to guide early prevention of stroke: identifying individuals at increased risk early in life, who would then be targeted for more intensive lifestyle modifications, similar to the roles that have been proposed for genetics in cancer risk stratification[38]. Unlike most established risk factors which may vary over time and are typically not informative at an early age, the metaGRS remains stable and can be derived from birth. Later in life, when measurements of established risk factors are available, these can be further combined with the metaGRS to give the most accurate prediction of a person's risk of incident stroke. Further research is required to determine what levels of risk factor reductions will be achievable and cost effective in practice.

Lastly, even for individuals within risk factor levels recommended by current guidelines (SBP < 120, BMI < 25, not currently smoking, no diagnosed diabetes), our models predict substantial differences in risk between different metaGRS levels. These results suggest that for individuals with high metaGRS, achieving currently recommended risk factor levels may not be sufficient and that it is time to contemplate whether future guidelines on primary and secondary stroke prevention should integrate genetic information when defining treatment goals for high-risk individuals. Ultimately, the practical implications of these results to stroke risk screening in the general population will require public health modelling, taking into account what is considered 'high risk' of stroke in the context of each country and health system, and the efficacy of interventions or treatments that are available for risk reduction.

Our study has several limitations. Compared with GRS for other common conditions, such as CAD[21], the performance of our metaGRS for stroke is limited. A likely reason is that stroke is more heterogeneous and that GWAS sample sizes for mechanistically defined stroke subtypes are still limited in comparison with other diseases. As stroke GWAS progress, GRS will become more powerful[39,40]. We did not observe substantial advantage from incorporating GRSs based on GWAS summary statistics for specific IS subtypes (LAS, CES, SVS) over that of IS as a

whole; however, we were not able to examine subtype-specific outcomes in the UKB due to the lack of such detailed information. There may still be benefit from developing subtype-specific scores that take advantage of the unique genetic architecture of each subtype[41]. The number of older individuals (>75 years) in UKB is limited at this stage, reducing our ability to model stroke risk in the age strata where the majority of events occur. Furthermore, the duration of follow-up in UKB is relatively limited and, because of the limited number of assessments, we could not model the cumulative effect of BP and smoking over time; however, we accounted for potential regression dilution bias in SBP measurements via the use of diagnosed hypertension, which showed stronger associations with stroke. Family history of stroke in UKB may be less comprehensive than in stroke-specific studies, limiting its predictive power, and overall the UKB study population is healthier than the general UK population[42], which could have led to under-estimation of some of the effects of risk factors. Further independent validation of the stroke metaGRS in other cohorts would be necessary before considering its clinical use; however, this is challenging given that the majority of available stroke GWAS studies have been included in MEGASTROKE. Nonetheless, recent successful validation of our previous CAD metaGRS[21] in French-Canadian cohorts[43] suggests that scores developed on the UK Biobank are generalisable to other cohorts of European ancestry. Our modelling assumes that risk factors, such as SBP and BMI, can be varied independently of each other. In practice, common lifestyle interventions such as exercise and diet will likely affect several risk factors at a time. Finally, this study focused on British white ancestry and it is yet unknown what performance is achievable in individuals of different ancestry;[44] successful development of scores in non-Europeans will require both stroke GWAS summary statistics from non-Europeans as well as sufficiently large prospective studies of stroke on which to validate these scores.

Taken together, despite challenges in phenotypic heterogeneity and corresponding GWAS power, our study presents the most powerful IS genomic risk score to date and assesses its potential for risk stratification in the context of established risk factors and clinical guidelines. It lays the groundwork for larger GWAS of stroke and its multiple subtypes as well as analyses which leverage the totality of information available for stroke genomic risk prediction.

## Methods
**Study participants**. This study was conducted under the UK Biobank project #26865, under the approval of the North West Multi-centre Research Ethics Committee (MREC) in the UK. All participants of UK Biobank provided written informed consent.

The UK Biobank (UKB) study[29,30] included individuals from the general UK population, aged between 40–69 years at recruitment. Recruitment included a standardised socio-demographic questionnaire, as well as medical history, family history, and other lifestyle factors. Several physical measurements (e.g., height, weight, waist-hip ratio, systolic and diastolic BP) were taken at assessment.

Individual records were linked to the Hospital Episode Statistics (HES) records and the national death and cancer registries. The age of event was age at the primary stroke event (the diagnostic algorithm for stroke in UKB can be found at http://biobank.ndph.ox.ac.uk/showcase/docs/alg_outcome_stroke.pdf; last accessed 11/04/2019).

We defined stroke risk factors at the first assessment, including: diabetes diagnosed by a doctor (field #2443), BMI (field #21001), current smoking (field #20116), hypertension, family history of stroke, and LDL cholesterol. For hypertension we used an expanded definition including self-reported high BP (either on BP medication, data fields #6177, #6153; or SBP > 140 mm Hg, fields #4080, #93; or DBP > 90 mm Hg, data fields #4079, #94) as well as hospital records; for registry cases, we use HESIN (hospital admission) and death registry data including both primary and secondary diagnoses/causes of death (HESIN: ICD9 401–405, ICD10 I10–I15; death: ICD10 I10–I15, data fields #40001, #40002). For family history of stroke, we considered history in any first-degree relative (father, mother, sibling; fields #20107, #20110, and #20111, respectively).

For individuals on BP-lowering medication, we adjusted SBP by adding +15 mm Hg as per Evangelou et al.[16,45]. We used LDL-cholesterol from the UKB biomarker panel, measured at first UKB assessment. For individuals on lipid-lowering medication at the time of assessment ($n = 66{,}737$), we adjusted the measured LDL-cholesterol level by +1.5 mmol L$^{-1}$.

We excluded individuals with withdrawn consent, self-reported stroke at age <20 years due the potential unreliability of these records, and those not of British white ancestry (identified via the UKB field 'in.white.British.ancestry.subset'[29]), leaving a total of $n = 407{,}388$ individuals. We censored the age of stroke at 75 years.

**Genotyping quality control**. The UKB v2 genotypes were genotyped on the UKB Axiom array, and imputed to the Haplotype Reference Consortium (HRC) by the UKB;[29] SNPs on the UK10K/1000Genomes panel were excluded from the current analysis. Imputed genotypes were converted to PLINK hard calls. For the initial GRS analysis, we considered genotyped or HRC-imputed SNPs with imputation INFO > 0.01 and global MAF > 0.001 (14.5 M autosomal SNPs). A further QC step was performed on the final metaGRS (see below).

**Generation of the metaGRS**. We randomly sampled $n = 11{,}995$ individuals from the UKB dataset, oversampling individuals with AS events, leading to 2065 individuals with AS (of which 889 were also IS events) and 9935 non-AS referents. This subset was used for developing GRSs, and was excluded from all further analysis. Five individuals were later removed due to withdrawn consent.

Using the UKB derivation set, we generated 19 GRSs for phenotypes associated with stroke (Supplementary Table 1). To minimise the risk of over-fitting due to overlap of individuals between the GWAS meta-analyses and the UKB validation dataset, we selected GWAS that did not include the UK Biobank in their meta-analysis.

The three CAD GRSs (46K, 1KGCAD, FDR202) were generated previously using an $n = 3000$ derivation subset of the UKB (included in the larger $n = 11{,}995$ subset employed here);[21] briefly: (i) the 46K score was derived by LD thinning of the Metabochip summary statistics;[46] (ii) the 1KGCAD was derived by LD thinning of the 1000Genomes CAD summary statistics;[47] and (iii) the FDR202 score was from the 1000Genomes CAD summary statistics, consisting of SNPs with associations at false discovery rate < 0.05. The AF GRS was derived from a GWAS of AF[48] using a pruning and thresholding approach. For the remaining GRSs, we used published summary statistics to generate a range of scores based on different $r^2$ thresholds with PLINK[49] LD thinning (–indep-pairwise), and selected one optimal model (in terms of the largest magnitude hazard ratio), resulting in one representative GRS for each set of summary statistics.

Each GRS was standardised (zero mean, unit standard deviation) over the entire dataset. Next, we employed elastic-net logistic regression[50] using the R package 'glmnet'[51] to model the associations between the 19 GRSs and stroke, adjusting for sex, genotyping chip (UKB vs BiLEVE), and 10 genetic PCs. A range of models with different penalties was evaluated using 10-fold cross-validation. The best model, in terms of highest cross-validated AUC (area under receiving-operating characteristic curve), was selected as the final model and held fixed for validation in the rest of the UKB data. The final adjusted coefficients for each GRS (odds ratios) in the penalised logistic regression are shown in Supplementary Fig. 1, in comparison with the univariate estimates (based on one GRS at a time).

The final per-GRS log odds $\gamma_1, \ldots, \gamma_{19}$ were converted to an equivalent per-SNP score via a weighted sum

$$\mathrm{GRS}_i^{\mathrm{meta}} \propto \sum_{j=1}^{m} x_{ij} \left( \frac{\gamma_1}{\sigma_1} \alpha_{j1} + \ldots + \frac{\gamma_{19}}{\sigma_{19}} \alpha_{j19} \right), \tag{1}$$

where $m$ is the total number of SNPs, $\sigma_1, \ldots, \sigma_{19}$ are the empirical standard deviations of each of the 19 GRSs in the derivation data, $\alpha_{j1}, \ldots, \alpha_{j19}$ are the SNP effect sizes (from the GWAS summary statistics) for the $j$th SNP in each of the GRSs, respectively, and $x_{ij}$ is the genotype {0, 1, 2} for the $i$th individual's $j$th SNP. A SNP's effect size $\alpha_{jk}$ was considered to be zero for the $k$th score if the SNP was not included in that score. This resulted in 3.6 million SNPs for inclusion in the metaGRS.

We conducted a sensitivity analysis to evaluate whether stricter quality control filtering would impact the performance of the metaGRS; removing SNPs with imputation INFO < 0.4 and MAF < 0.01 did not substantially affect the association of the metaGRS with stroke, hence, we selected the metaGRS with stricter QC as the final score, bringing the total number of SNPs to 3.2 million.

**Evaluation of the metaGRS**. The metaGRS developed using the derivation set was held fixed and evaluated in the UKB validation subset ($n = 395{,}393$) using a Cox proportional hazard model. We conducted complete case analysis due to the low proportion of participants with any missing values for the seven risk factor variables of interest (5.1% of participants).

Age was used as the time scale in the Cox proportional hazard regression. The regression was stratified by sex and weighted by the inverse probability of selection into the validation set, together with robust standard errors (R package 'survival'[52]). All analyses were adjusted for chip (UKB vs BiLEVE) and 10 PCs of the genotypes (as provided by UKB[29]). For analyses of incident stroke, age at UKB

assessment was taken as time of entry into the study. Cox models of the metaGRS did not show deviations from proportional hazard assumptions, based on the global test for scaled Schoenfeld residuals ($P = 0.32$).

The predicted cumulative risk curves (as a function of time $t$) were calculated using 'survfit.coxph' within each stratum of sex as

$$1 - \widehat{S}(t) = 1 - \exp\left(-\widehat{H}_0(t)\exp\left(\mathbf{x}^T\widehat{\boldsymbol{\beta}}\right)\right), \qquad (2)$$

where $\widehat{S}(t)$ is the cumulative survival at time $t$, $\widehat{H}_0(t)$ is the estimated baseline cumulative hazard at time $t$, $\mathbf{x}$ is the vector of the predictor variables set to the values of interest, and $\widehat{\boldsymbol{\beta}}$ is the vector of the estimated log hazard ratios for each predictor.

We performed a sensitivity analysis testing whether the association of the metaGRS with IS was affected by familial relatedness in the validation set. Relatedness analysis was done using KING[53] v2.1.4, based on ~784,000 autosomal SNPs measured on the Axiom chip, identifying $n = 336,643$ participants in the UKB validation set with kinship more distant than that of 2nd degree. There was a negligible difference in the association between the metaGRS and stroke in the full UKB validation set and within this distantly related subset of individuals.

Calibration of the metaGRS risk score was evaluated by fitting logistic regression models of the metaGRS (adjusting for sex, chip, and 10 genetic PCs) in the derivation set, predicting the absolute risk of event in the test set (allowing for the 9.38-fold lower observed baseline rate of events between the testing set compared with the derivation set), and evaluating the proportion of test set individuals with stroke events within each decile of the predicted risks (Supplementary Fig. 9). Pointwise confidence intervals were obtained via the binomial test for proportions.

We estimated the heritability of IS explained by the metaGRS, on the liability scale, using the $R^2$ and partial $R^2$ obtained from linear regression of the stroke outcomes on metaGRS (partial $R^2$ was from linear regression adjusted for sex, age of assessment, genotyping chip, and 10 PCs). The estimates were converted to the liability scale[54], assuming that the IS prevalence in UKB represents that of the general population ($K = 0.008$). Due to a lack of robust estimates of the heritability of stroke, we examined a range of plausible $h^2$ values from 0.1 to 0.4, yielding estimates of explained heritability ranging from 7.7 to 1.8%, respectively (Supplementary Fig. 10).

We performed sensitivity analysis to assess the effect of potential geographical stratification within the UKB[55] on the metaGRS. We compared the original metaGRS with residuals of the metaGRS regressed on (i) the first 10 PCs, (ii) first 10 PCs and natural cubic splines of the geographical north-south coordinates and east-west coordinates (3 degrees of freedom each), (iii) the first 30 PCs and splines of the coordinates, (iv) first 10 PCs and a thin-plate regression spline (TPRS) representing smooth interactions between the two coordinates[56], and finally (v) also adding the UKB assessment centre (Supplementary Fig. 11a). For the unadjusted score, we observed some variation across the north and east coordinates (up to 0.4 standard deviations), however, adjusting for PCs and the coordinates attenuated this variation substantially, with the TPRS method eliminating it completely. Despite the attenuation in geographical stratification, we observed negligible change in the association of the residuals of the scores with IS events (Supplementary Fig. 11b), indicating that any geographical stratification in UKB was not driving the metaGRS's association with stroke.

**Reporting summary.** Further information on research design is available in the Nature Research Reporting Summary linked to this article.

## Data availability

The SNP weights for the ischaemic stroke metaGRS are available at https://doi.org/10.6084/m9.figshare.8202233.

The UK Biobank genotype and phenotype data is available on application to the UK Biobank project at http://www.ukbiobank.ac.uk.

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

## Acknowledgements

This study was supported in part by the Victorian Government's OIS Program. M.I. was supported by an NHMRC and Australian Heart Foundation Career Development Fellowship (no. 1061435). G.A. was supported by an NHMRC Early Career Fellowship (no. 1090462). M.D. acknowledges funding from the European Union's Horizon 2020 research and innovation programme (grant agreements no. 666881, SVDs@target and no. 667375, CoSTREAM); and the DFG as part of the Munich Cluster for Systems Neurology (EXC 1010 SyNergy), the CRC 1123 (B3), and DI 722/13–1n. J.D. is funded by the National Institute for Health Research (Senior Investigator Award). M.I. and J.M.M.H. are funded by the National Institute for Health Research (Cambridge Biomedical Research Centre at the Cambridge University Hospitals NHS Foundation Trust). This work was supported by core funding from: the UK Medical Research Council (MR/L003120/1), the British Heart Foundation (RG/13/13/30194; RG/18/13/33946) and the National Institute for Health Research (Cambridge Biomedical Research Centre at the Cambridge University Hospitals NHS Foundation Trust). The views expressed are those of the authors and not necessarily those of the NHS, the NIHR, or the Department of Health and Social Care. The MEGASTROKE project received funding from sources specified at http://www.megastroke.org/acknowledgments.html. Data on coronary artery disease/myocardial infarction have been contributed by CARDIoGRAMplusC4D investigators and have been downloaded from www.cardiogramplusc4d.org. UK Biobank analyses were conducted under project 26865.

## Author contributions

G.A., M.I. and M.D. conceived and designed the study. G.A. carried out the statistical and computational analysis, with assistance from R.M., E.Y.-D., A.S. and T.W. The manuscript was written by G.A., M.I. and M.D. with contributions by all co-authors.

## Competing interests

Dr Butterworth has received grant support from Merck, Novartis, Pfizer, Biogen, Bioverativ, and AstraZeneca; and serves as a consultant to Novartis. The remaining authors declare no competing interests.
