## [Peer Review File · Nature Communications]

Reviewers' Comments:

Reviewer #1:

Remarks to the Author:

Gad Abraham and colleagues have conducted an interesting analysis generating a polygenic score for ischemic stroke. They pool multiple polygenic scores for stroke and related phenotypes in a derivation subset of UK Biobank to derive a metaGRS. They then validate the metaGRS in an independent sample. They demonstrate that this polygenic score can identify 0.25% of the population at three-fold risk and that this score has similar predictive power to conventional risk factors.

The analysis is well conducted and represents an interesting advance that could be applied to other phenotypes. I have minor comments below.

Comments:

1. Derivation of the score in a random sample of UK Biobank is likely to result in overperformance of the score in a validation set of UK Biobank. Can the authors replicate the performance of the score in a truly independent sample?
2. Derivation of a metaGRS using genomic information from multiple related phenotypes such as CAD and risk factors is, in my view, an interesting idea that could be applied to polygenic prediction of other phenotypes. I would highlight this approach and the improved performance resulting from it in the abstract.
3. One thing that is not clear is whether inclusion of the polygenic scores for coronary artery disease and for risk factors improved the metaGRS. Could the authors add an analysis examining whether the CAD and risk factors improved the performance of the score eg into supp figure 2?
4. The actual performance of the score is overstated. Identifying 0.25% of the population at three-fold risk is far inferior to polygenic scores for other diseases such as coronary artery disease or breast cancer. For example, a polygenic score for CAD can identify 8% of the population at three fold risk. The top 1% of the score is at >5 fold risk (Khera et al.).
5. What was the risk of any stroke in the top 0.25% of the population? There appears to be a significant drop off in the power of the score to predict any stroke, despite the authors reporting that ischemic stroke represents 80% of strokes.

Reviewer #2:

Remarks to the Author:

In the manuscript "Genomic risk score offers predictive performance comparable to clinical risk factors for ischaemic stroke", Abraham et al. describe the beginnings of the future model of stroke risk stratification and therefore stroke prevention based on a combined genomic risk score. They tested the association between multiple GRS scores, combined into a single metaGRS, and ischemic stroke derived and tested on the UK biobank. They further compared predictive performance of the genomic model to long standing conventional risk factor models. Overall, I believe this manuscript provides a significant advance in the collective field of stroke prevention and deserves publication in Nature communications.

The manuscript is well written and a good read. The derivation of the 19 GRSs is random in some instances (inclusion of 3 CAD models), but appropriate given that the goal is better prediction. The statistical analysis of highly correlated predictors using elastic net regression is appropriate but over-fitting is a concern which is appropriately mentioned. The validation phase using a separate cohort is standard but in the "generation of the metaGRS" section the following comment is confusing - "inclusion of the same individuals in the derivation and validation datastes.." and should be clarified.

The study found that the metaGRS has predictive value in ischemic stroke in total, which is similar to that of hypertension alone. A model including all conventional risk factors, outperformed the metGRS group but a combination of the two had some cumulative value.

Comments:

The Abstract is a bit misleading: The authors state "the metaGRS was similarly or more predictive when compared to established risk factors...". Based on Figure 4, MetaGRS is not as good of a predictor of stroke compared to "hypertension" and is far inferior to "all risk factors combined". Further, the value of the MetaGRS added to the conventional risk factors model, was minor – something which was not mentioned. The findings of the study should be better and more accurately summarized in the abstract.

While the authors discuss the stroke subtypes as a potential limitation of GRS-based prediction models in the intro, they did not actually describe predictive value of their models by subtype. If possible, the output of the prediction phase, should be broken down by subtype or at the very least, a comment should be made as to why this was not done.

The validation cohort should be independent of the derivation cohort (see above)

It is not clear if the model was validated on patients without a history of stroke and thus a risk stratification model in stroke-naïve patients. If there were patient with a history of stroke included, stroke should be added as a risk factor in the conventional models or a mention should be made.

In the second paragraph of the discussion: while an association here is not the same as predictive value, positive or negative, the group should try to comment on what this means if eventually adopted in clinical practice. For example, applying the metaGRS alone identified 1 in x high risk individuals, and found an additional 1 in x additional high risk individuals when combined with conventional stroke risk factors, compared to conventional stroke risk factors alone. To their credit, they attempted to do that in the second paragraph of the discussion but their comparison is somewhat difficult to understand. They mention that 1 in 400 individuals are predicted as high risk, but didn't comment on how many people weren't perceived as high risk but still had stroke, nor did they comment on the additional predictive benefit of the model, compared to the conventional risk factors alone. Much of this is summarized in figure 5. The language used in the "predicting ischaemic stroke risk with established risk factors and the metaGRS" section is very helpful in understanding this real world application and should be used in the discussion; in addition to or in place of the current wording.

Finally, the authors recognized the limitation that the cohort in which this model was developed was made up of white patients of British ancestry. The importance of diversity, both ethnic and racial, in early genetic studies cannot simply be a "by the way" statement. Given the future implications, the authors cannot understate the implications to reproducibility and the potential for furthering racial/ stroke disparities.

Reply to reviewer comments

Reviewer #1

Minor comments

1. Derivation of the score in a random sample of UK Biobank is likely to result in overperformance of the score in a validation set of UK Biobank. Can the authors replicate the performance of the score in a truly independent sample?

Reply: We thank the reviewer for raising this point. The concern is not entirely clear to us: is it about (i) statistical *over-fitting* per-se or (ii) about having a *UK Biobank-unique* score that may not generalise to other cohorts due to some special characteristics of the UK Biobank (ethnicity, age, etc)? We address each concern below.

a. Statistical overfitting: We respectfully disagree that random splitting of the UK Biobank samples into a tuning and validation set will result in over-fitting of the score. Our score was developed on the tuning set, then held *fixed*, and tested in the validation set; there is no overlap of individuals between the two subsets (also see reply to Reviewer 2, point #3). This procedure is the same as hold-out validation, which is a widely accepted method for controlling for overfitting.

To further demonstrate the validity of this procedure within the same data, we have performed an analysis comparing our metaGRS approach (model development on the tuning set and testing on the validation set) with an alternative approach, *smtPred* (Maier et al 2018, *Nat Comms* PMID 29515099), using four component GRSs (Any Stroke [AS], Ischaemic Stroke [IS], Body Mass Index [BMI], and Systolic Blood Pressure [SBP]) with essentially identical results (see **Figure 1** below). *smtPred* is a multi-trait prediction method which uses the estimated heritabilities and genetic correlations between traits (obtained from LD Score Regression) to weight the GRSs; these parameters are estimated from summary statistics alone and are independent of the UK Biobank. **This analysis indicates that our metaGRS procedure (employing the UKB derivation/validation split) has the same performance as *smtPred* and thus does not lead to any over-fitting.** Importantly, however, we cannot perform the *smtPred* analysis for all 19 GRSs used in the manuscript since not all of them are suitable for LD Score Regression as some are not based on genome-wide summary statistics (i.e., the AF, FDR202, and GRS46K GRSs). Nonetheless, the 4-score metaGRS was derived using the same approach as the full 19-score metaGRS, hence we have no reason to believe the full score suffers from overfitting either.

b. Our approach externally validates for other related GRSs: Identical approaches have successfully been used on the UK Biobank before: in Inouye et al 2018 (*JACC*, PMID 30309464) we derived a coronary artery disease GRS (metaGRS_{CAD}), and used n=3000 as a tuning set and validated on the rest. Khera et al 2018 (*Nat Genet*, PMID 30104762) used a larger subset of the UK Biobank for tuning (n=120,000) and tested on the rest. Importantly, the metaGRS_{CAD} has been successfully externally validated in three French-Canadian cohorts by Wünnemann et al 2019 (*Circulation GPM*, PMID 31184202) and in the Mayo Vascular Disease Biorepository (Dikilitas et al 2019, DOI:10.1016/S0735-1097(19)32399-X). In meta-analysis across the three French-Canadian cohorts the metaGRS_{CAD} had an odds ratio OR=1.69 per standard deviation (95% CI 1.58–1.81), very close to the result of OR=1.76 per standard deviation (95% CI 1.73–1.78) in Inouye et al. In the Mayo dataset, the metaGRS_{CAD} had an odds ratio of 2.30 (95% CI 1.93–2.74) for the top 20% of the participants vs the rest, which is again consistent with the performance we observed in the UK Biobank (OR=2.56, 95% CI 2.48–2.64 for the top 20% vs the rest).

c. External validation of our stroke metaGRS would be challenging and possibly lead to over-fitting. The majority of European ancestry ischaemic stroke cohorts in the world (e.g., NINDS-SiGN, CHARGE) have been included in the MEGASTROKE meta-analysis (from which the GWAS summary statistics are derived). Hence, ‘validation’ in these samples would not be a truly independent validation, and could lead to inflation of the apparent predictive power. Second, cohorts that were not part of the MEGASTROKE meta-analysis (e.g., the Million Veterans Program; MVP) would be difficult and time-consuming to obtain access to. In any

case, the MVP is arguably even less representative of the general population than the UK Biobank, being based on predominantly male American veterans with relatively high rates of heart failure and CVD, rather than a random sample of the United States population.

Reply Figure 1: Comparison of the performance of a metaGRS based on four component GRSs (Any Stroke [AS], Ischaemic Stroke [IS], Body Mass Index [BMI], and Systolic Blood Pressure [SBP]), an smtPred model based on the same four GRSs, and a single GRS for IS. The metaGRS was tuned in 10-fold cross-validation on the n=11,995 tuning set. Results are for the UK Biobank validation set (n=395,393).

We have made the following changes:

- *Results:* Added a description of the above smtPred analysis to the manuscript (pg 3): “We performed a sensitivity analysis to assess whether the estimation of the metaGRS weights on the UKB derivation set led to over-fitting (upwards bias in apparent performance) of the score in the validation set. We developed a metaGRS based on four component GRSs (Any Stroke [AS], Ischaemic Stroke [IS], Body Mass Index [BMI], and Systolic Blood Pressure [SBP]) in cross-validation on the derivation set. We compared this metaGRS with a score derived using smtPred²⁸, which relies on the chip heritabilities and genetic correlations estimated from the GWAS summary statistics via LD score regression^{31,32}, independently of the UKB (Supplementary Figure 2). Overall, the two scores were highly correlated (Pearson $r=0.98$), and had indistinguishable associations with ischaemic stroke in the UKB validation set, indicating that our metaGRS procedure did not lead to overfitting in the validation set.”
- *Supplementary Results:* Added **Reply Figure 1** as Supplementary Figure 2.
- *Discussion* (limitations paragraph, middle of pg 7): Added the text “Further independent validation of the stroke metaGRS in other cohorts would be necessary before considering its clinical use; however, currently, validation is challenging given that the majority of available stroke GWAS studies have been included in MEGASTROKE. Nonetheless, recent successful validation of our previous CAD metaGRS²¹ in French-Canadian cohorts⁴³ suggests that scores developed on the UK Biobank are generalisable to other cohorts of European ancestry”.

2. Derivation of a metaGRS using genomic information from multiple related phenotypes such as CAD and risk factors is, in my view, an interesting idea that could be applied to polygenic prediction of other phenotypes. I would highlight this approach and the improved performance resulting from it in the abstract.

Reply: We agree with the reviewer about the potential of this method, particularly as it is challenging and time-consuming to substantially increase GWAS sample sizes.

We have made the following changes:

- **Abstract:** Added the highlighted text “The metaGRS hazard ratio for IS ... is stronger than an IS-only score (HR=1.18, 95 %CI 1.15–1.22)”. (Note: we have shortened the original Abstract).
- **Results (pg 3):** Added the highlighted text “Using the independent UKB validation set ... The metaGRS was associated with IS with a HR of 1.26 (95% CI 1.22–1.31) per standard deviation of metaGRS, which was stronger than any individual GRS comprising the metaGRS (including the IS GRS [HR=1.18, 95% CI 1.15–1.22]) and was twice the effect size of the previously published 90-SNP IS score²⁴ (HR=1.13 [95% CI 1.10–1.17]; Supplementary Figure 3a)”.

3. One thing that is not clear is whether inclusion of the polygenic scores for coronary artery disease and for risk factors improved the metaGRS. Could the authors add an analysis examining whether the CAD and risk factors improved the performance of the score eg into supp figure 2?

Reply: We thank the reviewer for raising this. Supplementary Figure 1 shows that the contributions to the metaGRS (apart from the AS and IS scores) were (in descending order of magnitude of the log odds ratio) from: AF, BMI, FDR202, DBP, TC, HDL, Height, LAS (an IS subtype), SVS (an IS subtype), and finally the GRS46K. Following the reviewer’s suggestion we have done another analysis comparing four GRSs (see **Reply Figure 2** below):

- the IS-only GRS;
- a metaGRS based on stroke-related GRSs (AS, IS, CES, LAS, SVS) and CAD-related GRSs (46K, FDR202, 1KGCAD) but no other risk factors;
- a metaGRS based on stroke-related GRSs (AS, IS, CES, LAS, SVS) and risk-factor GRSs (SBP, DBP, TC, LDL, HDL, TG, AF, BMI, Height, T2D, Smoking) but no CAD-related GRSs;
- the original 19-component metaGRS.

Figure 2 below shows that the addition of either CAD-related GRS or risk-factor-related GRS to the score had incremental impact on performance. Together with the odds ratios discussed above, these results indicate that the GRSs for CAD and the risk factors each have an impact on the final performance, but it is difficult to precisely tease out the relative contribution of each class of GRS, partly due to limitations in sample size of the UKB (leading to imprecise effect size estimates) and partly because there is overlap between GRSs due to pleiotropy; e.g., CAD GRSs themselves contain some information about risk factors such as blood pressure and cholesterol since these are risk factors for CAD as well as for stroke.

Reply Figure 2: Performance of four GRSs/metaGRSs in the UKB validation dataset (n=395,393). See above for description of each score. *RF*: stroke risk factors.

We have made the following changes to the manuscript:

- *Results:* We have added a discussion of this analysis to the manuscript (pg 4): “To further assess the contribution of different classes of GRS in the final score, we constructed two metaGRSs: (i) a score based on stroke-related GRSs (AS, IS, CES, LAS, SVS) and CAD-related GRSs (46K, FDR202, 1KGCAD) but no other risk factors; and (ii) a metaGRS based on stroke-related GRSs (AS, IS, CES, LAS, SVS) and risk-factor-related GRSs (SBP, DBP, TC, LDL, HDL, TG, AF, BMI, Height, T2D, Smoking) but no CAD-related GRSs (Supplementary Figure 6). Addition of either risk-factor GRSs or CAD-related GRSs each led to more powerful metaGRSs compared with the IS-only GRS, but the best score was achieved when combining both types of GRS into the 19-component metaGRS, indicating that both types of GRS had independent information about stroke risk. Note that due to pleiotropy there is some overlap between the genetic signal for CAD and risk factors such as blood pressure and cholesterol.”
- *Supplementary Results:* We have added the above **Reply Figure 2** as Supplementary Figure 6 to the manuscript.

4. The actual performance of the score is overstated. Identifying 0.25% of the population at three-fold risk is far inferior to polygenic scores for other diseases such as coronary artery disease or breast cancer. For example, a polygenic score for CAD can identify 8% of the population at three fold risk. The top 1% of the score is at >5 fold risk (Khera et al.).

Reply: We apologise for giving the reviewer the impression that there is overstatement. We of course agree with the reviewer regarding the performance of our score compared to coronary artery disease and breast cancer; however, our intention was to compare the risk and frequency of the ischaemic stroke metaGRS with monogenic forms of stroke. There is not a stark increase of frequency for a given risk as compared to CAD or breast cancer, but we note that a likely reason is that stroke is a more heterogeneous phenotype and that GWAS sample sizes for mechanistically defined stroke subtypes (e.g. large artery stroke, small vessel stroke, cardioembolic stroke) are still rather limited in comparison with other diseases. Nevertheless, we emphasise that our study presents by far the strongest polygenic risk score for ischaemic stroke at present.

We have made the following changes to the manuscript:

- *Discussion:* We now expanded on the limitations paragraph (pg 7): “Compared to GRS for other common conditions, such as CAD²¹, the performance of our metaGRS for stroke is limited. A likely reason is that stroke is more heterogeneous and that GWAS sample sizes for mechanistically defined stroke subtypes are still limited in comparison with other diseases. As stroke GWAS progress, genomic risk scores will become more powerful^{39,40”}

5. What was the risk of any stroke in the top 0.25% of the population? There appears to be a significant drop off in the power of the score to predict any stroke, despite the authors reporting that ischemic stroke represents 80% of strokes.

Reply: The hazard ratio for any stroke for the 0.25% of the population vs the middle 45–55% was HR=1.98 (95% CI 1.42–2.75), which indeed is lower than for IS but still substantial. This drop off in predictive power likely relates to differences in the genetic architecture and mechanisms underlying ischaemic stroke (about 80% of all stroke cases) and haemorrhagic stroke (about 20% of all stroke cases). In fact, substantial differences in the genetic architecture between the two phenotypes imply that a GRS for ischaemic stroke will likely not be as predictive for haemorrhagic stroke or for any stroke (as we have observed in practice, see reply to Reviewer 2, #2). To rigorously assess the reasons for the observed drop-off in predictive power, or indeed to correct it, would require a large GWAS of haemorrhagic stroke, which is currently not available. Of note, in the latest MEGASTROKE meta-analysis (Malik et al, 2018, PMID 29531354), 90% of the stroke cases were ischaemic and the number of loci reaching genome-wide significance in MEGASTROKE was, as expected, smaller for any stroke than for ischaemic stroke alone.

Reviewer #2

1. The Abstract is a bit misleading: The authors state “the metaGRS was similarly or more predictive when compared to established risk factors...”. Based on Figure 4, MetaGRS is not as good of a predictor of stroke compared to “hypertension” and is far inferior to “all risk factors combined”. Further, the value of the MetaGRS added to the conventional risk factors model, was minor – something which was not mentioned. The findings of the study should be better and more accurately summarized in the abstract.

Reply: We do not wish to ‘oversell’ the predictive power of the metaGRS, but at the same time it is important to highlight the result that the GRS for stroke was more predictive than a number of established risk factors, including family history, BMI, current smoking, diagnosed diabetes, and was essentially the same as systolic BP. While hypertension was indeed stronger, we used a wide definition for hypertension, based on a combination of BP at UKB assessment, self-reported high BP, BP medication usage, and hospital and death registry data (primary and secondary causes of death).

We have made the following changes to the manuscript:

- *Abstract:* We have modified the text to add the word several to “The metaGRS was similarly or more predictive when compared to several established risk factors, such as family history, blood pressure, body mass index, and smoking status.” (Note: we have shortened the original Abstract in order to comply with Nature Communications editorial policies).

2. While the authors discuss the stroke subtypes as a potential limitation of GRS-based prediction models in the intro, they did not actually describe predictive value of their models by subtype. If possible, the output of the prediction phase, should be broken down by subtype or at the very least, a comment should be made as to why this was not done.

Reply: Unfortunately, the UKB stroke outcomes available to us only included information about any stroke, ischaemic stroke, and two subtypes of haemorrhagic stroke (subarachnoid [n=821 events] and intracerebral [n=639 events]), limiting our ability to examine ischaemic stroke subtypes. As expected, the metaGRS was more weakly associated with subarachnoid and intracerebral stroke than with ischaemic stroke (HR=1.09, 95% CI 1.02–1.17 and HR=1.09, 95% CI 1.01–1.18, respectively). As we note above, an analysis that focuses on stroke subtypes is underpowered compared to ischaemic stroke as a whole and the relatively low case numbers in stroke GWAS.

We have made the following changes to the manuscript (highlighted by underscore):

- *Discussion* (limitations paragraph, pg 7): Modified the text to say “We did not observe substantial advantage from incorporating GRSs based on GWAS summary statistics for specific IS subtypes (LAS, CES, SVS) over that of IS as a whole; however we were not able to examine subtype-specific outcomes in the UKB due to the lack of such detailed information. There may still be benefit from developing subtype-specific scores that take advantage of the unique genetic architecture of each subtype”.

3. The validation phase using a separate cohort is standard but in the “generation of the metaGRS” section the following comment is confusing - “inclusion of the same individuals in the derivation and validation datasets..” and should be clarified ... the validation cohort should be independent of the derivation cohort.

Reply: We regret this text was not clear, resulting in some confusion. In our analysis, there was **absolutely no overlap between the derivation and validation sets** (n=11,195 and n=395,393, respectively). This may have been poor wording on our part: the point we were making was about the independence of the *GWAS summary statistics from the UK Biobank rather than the UK Biobank derivation/validation split*: we purposefully selected summary statistics that were not based (wholly or in part) on the UK Biobank in order to avoid any possibility that the validation set would contain some of the individuals used for the GWAS meta-analysis.

We have made the following changes to the manuscript:

- *Methods:* We have revised the text to make this point more clearly (pg 9): “To minimise the risk of over-fitting due to overlap of individuals between the GWAS meta-analyses and the UKB validation dataset, we selected GWAS that did not include the UK Biobank in their meta-analysis.”

4. It is not clear if the model was validated on patients without a history of stroke and thus a risk stratification model in stroke-naïve patients. If there were patient with a history of stroke included, stroke should be added as a risk factor in the conventional models or a mention should be made.

Reply: All ischaemic stroke end points were defined as primary events, i.e., the first reported stroke event. For the analyses of prevalent + incident stroke (Figure 3 in the manuscript), this includes all individuals in the validation dataset (n=395,393 in total). For analyses of incident stroke only (Figure 4 in the manuscript), i.e., those individuals either (i) without stroke or (ii) with stroke recorded *after* the UKB baseline assessment (n=390,849 in total), by necessity we excluded individuals with pre-existing ischaemic stroke (prior to baseline assessment). We did not consider prediction of secondary ischaemic stroke events as the clinical pathways for these individuals would be very different from individuals who are at increased risk but have not yet had a stroke.

There were a small number of individuals who are recorded as having experienced haemorrhagic stroke prior to ischaemic stroke (n=45 out of 3075 ischaemic stroke events). We conducted a sensitivity analysis to examine the impact of this on the results: (i) removing all n=45 events from the Cox regression and (ii) adjusting for the prevalent haemorrhagic stroke status. In both cases, there was essentially no change to the association of the metaGRS with ischaemic stroke (HR=1.27 per standard deviation of the metaGRS, in the two sensitivity analyses vs HR=1.26 in the original analysis).

We have made the following changes to the manuscript:

- *Results:* We have added text to the manuscript (pg 4): “A small number of individuals (n=45) had recorded haemorrhagic stroke before their primary ischaemic stroke event. We conducted two sensitivity analyses to assess the impact of this on our results: (i) excluding n=45 individuals from the analysis; (ii) adjusting for haemorrhagic stroke status in the analysis. In both cases, there was essentially no difference in the association of the metaGRS with ischaemic stroke compared with the original analysis (HR=1.27 per standard deviation of the metaGRS across the two analyses).”

5. In the second paragraph of the discussion: while an association here is not the same as predictive value, positive or negative, the group should try to comment on what this means if eventually adopted in clinical practice. For example, applying the metaGRS alone identified 1 in x high risk individuals, and found an additional 1 in x additional high risk individuals when combined with conventional stroke risk factors, compared to conventional stroke risk factors alone. To their credit, they attempted to do that in the second paragraph of the discussion but their comparison is somewhat difficult to understand. They mention that 1 in 400 individuals are predicted as high risk, but didn't comment on how many people weren't perceived as high risk but still had stroke, nor did they comment on the additional predictive benefit of the model, compared to the conventional risk factors alone. Much of this is summarized in figure 5. The language used in the “predicting ischaemic stroke risk with established risk factors and the metaGRS” section is very helpful in understanding this real world application and should be used in the discussion; in addition to or in place of the current wording.

Reply: We very much agree with the reviewer as to the importance of quantifying the benefit of adding the IS metaGRS to conventional risk factors in terms of identifying high-risk individuals, especially those who would potentially not have been identified using existing risk factors alone.

Our existing analysis partly answers this question by predicting the absolute risk for each individual and showing that the absolute risk can vary quite substantially by metaGRS bins (2.3-fold) even for those individuals who would normally be classified as within the risk factor guidelines (Figure 5b). However, we would caution against prematurely making claims like ‘1 in x high-risk individuals can now be identified’ because these numbers would critically depend on (i) having an agreed definition

of what is 'high risk' of stroke, which can vary between countries and health systems as well as strongly depends on age; (ii) being able to extrapolate from the UKB to the general UK population, which is challenging because the benefit in the UKB would likely under-estimate the true benefit in the general UK (fewer people would be considered 'high risk' in the UKB, making the results seem less impactful than they really are). For these reasons, we believe that this issue requires a proper public health analysis and is beyond the scope of the current paper and best left to a follow-up paper that can address this important issue more comprehensively.

We have made the following changes to the manuscript:

- *Discussion* (bottom of pg 6): We have added the text to the paragraph beginning with "Lastly, even for individuals", "Ultimately, the practical implications of these results to stroke risk screening in the general population will require public health modelling, taking into account what is considered 'high risk' of stroke in the context of each country and health system, and the efficacy of interventions or treatments that are available for risk reduction".

6. Finally, the authors recognized the limitation that the cohort in which this model was developed was made up of white patients of British ancestry. The importance of diversity, both ethnic and racial, in early genetic studies cannot simply be a "by the way" statement. Given the future implications, the authors cannot understate the implications to reproducibility and the potential for furthering racial/ stroke disparities.

Reply: We strongly agree with the reviewer about the importance of this issue. This is unfortunately a challenging issue bigger than one paper, as it is a limitation of both the GWAS meta-analyses (which are mostly in European individuals) and the UK Biobank itself (which consists mostly of British white individuals), which has to be addressed an international level through increased funding and awareness.

We have made the following changes:

- *Discussion* (limitations paragraph, middle of pg 7): We have modified the text in the penultimate paragraph to expand on this point "Finally, this study focused on British white ancestry and it is yet unknown what performance is achievable in individuals of different ancestry⁴⁴; successful development of scores in non-Europeans will require both stroke GWAS summary statistics from non-Europeans as well as sufficiently large prospective studies of stroke on which to validate these scores."

Reviewers' Comments:

Reviewer #1:

Remarks to the Author:

The authors have appropriately addressed my concerns.

Reviewer #2:

None